# Co-Occurrence of Domestic Dogs and Gastropod Molluscs in Public Dog-Walking Spaces and Implications for Infection with *Angiostrongylus vasorum*: A Preliminary Study

**DOI:** 10.3390/ani11092577

**Published:** 2021-09-02

**Authors:** Bryony A. Tolhurst, Andrew D. J. Overall, Peter J. King, Eric R. Morgan, Rowenna J. Baker

**Affiliations:** 1Ecology, Conservation and Zoonosis Research and Enterprise Group, The University of Brighton, Huxley Building, Lewes Road, Brighton BN2 4GJ, UK; A.D.J.Overall@brighton.ac.uk (A.D.J.O.); row_baker@hotmail.com (R.J.B.); 2Independent Researcher, Storrington RH20 4NZ, UK; peter.king@oart.org.uk; 3Biological Sciences, Queens University Belfast, 19, Chlorine Gardens, Belfast BT9 5DL, UK; Eric.morgan@qub.ac.uk

**Keywords:** slug, snail, gastropod mollusc, *Angiostronglyus vasorum*, domestic dog, *Canis lupus familiaris*, co-occurrence, intermediate host, spatiotemporal overlap, habitat, urban ecology

## Abstract

**Simple Summary:**

*Angiostrongylus vasorum* is a serious parasitic disease increasing in range and prevalence in Europe. The parasite passes through land slugs and snails before it can infect dogs but contact between dogs and these intermediate hosts is not well studied. We surveyed dogs and slugs/snails in parks and on streets in an urban *A. vasorum* hotspot area in southern England, United Kingdom, with the aim of determining the conditions under which they overlap. We counted 1672 slugs/snails and 763 dogs across seven sites. We found that habitat types in which the hosts were present differed, with dogs occurring 15× more often on hard surfaces (e.g., concrete) than woodland/scrub, but also occurring on natural grassland. Large numbers of slugs/snails were present 5.82× more often in woodland/scrub and natural grassland than on hard surfaces. Slug and snail species at risk of a greater likelihood of infection with *A. vasorum* were present 65.12× more often in woodland/scrub and 62.17× more often in amenity grassland than other habitats. The results suggest that contact between dogs and slugs/snails is most likely in amenity and natural grassland but that infection risk with *A. vasorum* is greatest in amenity grassland and woodland/scrub.

**Abstract:**

*Angiostrongylus**vasorum* is a helminth parasite of domestic dogs that is increasing in range and prevalence. Its lifecycle requires terrestrial gastropod mollusc (“gastropod”) intermediate hosts, but research is lacking regarding contact risk in situ. We studied co-occurrence between dogs and gastropods in dog-walking spaces in an *A. vasorum* hotspot in southern England, United Kingdom, with the aim of quantifying environmental and spatio-temporal overlap. We surveyed 390 quadrats and 180 point-counts along 3 km transects at seven sites, yielding 1672 gastropod and 763 dog observations. Common gastropods comprised *Arion*, *Cornu*, *Monacha*, *Deroceras*, *Tandonia*, *Cochlicella,* and *Trochulus* species. Habitat was the most important factor structuring both gastropod and dog presence and abundance. Likelihood ratio comparisons from conditional probability trees revealed that dogs were 15× more likely to be present on hardstanding surfaces than other habitats but were also present on natural and amenity grassland. Presence of gastropod species associated with high *A. vasorum* prevalence was 65.12× more likely in woodland/scrub and 62.17× more likely in amenity grassland than other habitats. For gastropods overall, high abundance was 5.82× more likely in woodland/scrub and natural grassland. The findings suggest co-occurrence is highest in amenity and natural grassland, but infection risk is greatest in amenity grassland and woodland/scrub.

## 1. Introduction

Prevalence and geographical distribution of canine angiostrongylosis caused by infection with the nematode parasite *Angiostrongylus vasorum* are increasing globally [1]. Originally unevenly distributed across the United Kingdom (UK) and Ireland, *A. vasorum* is now widespread in red foxes (*Vulpes vulpes*) [2] and a serious threat to the health of domestic dogs (*Canis lupus familiaris*) in endemic areas, particularly in southern England [3]. This species is the most pathogenic of the lungworms in dogs [4] with a mortality rate irrespective of treatment ranging from 2 to 13% in referred cases [5,6]. Prevalence in dogs in the UK is below 4% overall [7], but above 16% in dogs with consistent clinical signs, which are highly variable [8,9].

Completion of the *A. vasorum* lifecycle is indirect, requiring larval development in intermediate hosts, normally terrestrial gastropod molluscs (hereafter termed “gastropods”) but potentially also amphibians [10]. Amphibians and birds can additionally act as paratenic hosts [11]. UK studies of natural infection in gastropods have largely focused on slugs of the *Arion hortensis* and *Arion ater* species aggregates (morphologically indistinct and inter-breeding species comprising *A. hortensis, Arion distinctus* and *Arion owenii*; and *A. ater, Arion rufus* and *Arion vulgaris* = *Arion lusitanicus,* respectively [12]) and the snail *Cornu aspersum*, as prominent intermediate hosts [4,9,13,14]. However, L3 (infective stage) *A. vasorum* larvae have also been isolated from other gastropod species including slugs in the genera *Derocerus, Tandonia* and *Limax* [9,15,16] while the lack of intermediate host specificity of metastrongyloid parasites suggests that the *A. vasorum* host range is larger still. Dogs are most likely to become infected by either intentionally or accidentally ingesting infected gastropods [4]. Although *A. vasorum* larvae have additionally been isolated from gastropod faeces [17] and from water holding experimentally infected aquatic snail hosts [18], the epidemiological significance of these potential routes of infection is unknown [17].

Management of canine angiostrongylosis is currently largely focused on prophylactic medication and remedial treatment, consisting of the anthelmintics moxidectin or milbemycin oxime [4,19,20]. Prevention or reduction of *A. vasorum* transmission via behavioral means has not been explored empirically, although [4] recommend common-sense measures of intermediate host avoidance by dog walkers based on gastropod mollusc activity patterns. Detailed knowledge of natural infection cycles in intermediate hosts is key for predicting risk of angiostrongylosis in dogs [13] and behavioral correlates of infection cycles correspondingly require investigation. Further, *A. vasorum* prevalence in gastropod populations is under-studied but believed to be highly geographically variable in the UK; for example, using polymerase chain reaction (PCR), 0.3% of sampled gastropods were infected in Bristol, 4% in Guildford, 6% in Glasgow, and 29% in Swansea [9,14,16], although differences in study design and timing mean that these figures are not comparable. The large, coprophilic arionid slugs were recorded to be the most commonly infected [14] and greater larval burdens have been recorded in larger, heavier slugs [13,21]. Despite this, dogs are most likely to ingest smaller gastropods incidentally, and these species may be underestimated by traditional, nocturnal gastropod surveys due to their lower detection rate. Over-dispersion of larval burdens in infected slugs has been reported [21] potentially creating asymmetry in infection risk per given contact between dogs and individual gastropods. Certainly, variation between gastropod species in terms of activity in situ, *A. vasorum* prevalence and aggregation, and the relative likelihood of contact and ingestion by dogs, remain poorly understood.

Some of the variation in observed *A. vasorum* prevalence in sampled slugs is also likely to arise from seasonal differences and landscape type, with higher rates of infection recorded in Autumn than Summer and in semi-urban environments versus rural areas [14]. Abundance, species composition and diversity of gastropod species in the UK is strongly influenced by climate, and conditions for native species are predicted to shift rapidly in response to climate change [22], and potentially release from temperature-sensitive parasites [23]. Gastropod community dynamics and transmission of *A. vasorum* could further be influenced by invasion of exotic slugs. For example, the ‘Spanish slug’ (*A. vulgaris* = *A. lusitanicus*), the best-studied species implicated in the *A. vasorum* system in Europe, is strongly affected by humidity [24,25,26,27] and temperature [27,28,29] and shows phenotypic plasticity that aids invasion [30]. The activity (feeding and locomotion) of this species is also affected by time of day, peaking approximately an hour after sunrise and sunset [31]. Changes in gastropod species composition, abundance and activity are therefore highly likely to alter the effects of climate on *A. vasorum* transmission [21,32] and might help explain the rapid emergence of this parasite in the UK [2] and other European countries (e.g., [33,34]).

Few studies have investigated the *A. vasorum* system in situ, beyond assessment of prevalence in gastropod and/or dog populations. Further, mechanistic studies based on host behavior are almost entirely absent from the *A. vasorum* literature. Therefore, the interplay between gastropod habitat selection, abundance and species composition, dog abundance and habitat selection and climatic variables remains largely unexplored. There is also a paucity of evidence-based guidance for dog owners as to how they can lower the risk of contact between their dogs and gastropods, thus potentially reducing risk of infection.

We sought to redress this by simultaneously collecting field data on the occurrence of domestic dogs and gastropods in an urban/peri-urban area in southern England, UK, to estimate the probability of co-occurrence and thus potential *A. vasorum* transmission, in representative public dog-walking spaces. Our aim was to identify patterns of the probability of co-occurrence *P(co*) between dogs and gastropods overall, and between dogs and gastropod species in which *A. vasorum* is known to occur at medium or high prevalence from the literature, under varying in situ conditions. In doing so, we aimed to identify spatiotemporal foci of risk for contact, and potentially, infection [*P(inf)*]. We expected gastropod occurrence to peak in warm wet conditions (i.e., 13–15 °C and 80–100% relative humidity) based on mean activity peaks from previous unpublished observations and in the early morning and late evening. Although dog owners might be expected to walk their dogs more often under clement conditions, there is limited evidence from a small body of literature [35] that weather affects dog walking behavior. Our findings provide information for dog owners as to differential risk of dog walking under different spatiotemporal scenarios with the ultimate aim of broadening the toolkit available to them to protect their dogs from angiostrongylosis.

## 2. Materials and Methods

### 2.1. Study Area

The study took place within the city of Brighton and Hove in East Sussex, England, UK (Latitude = 50.82253, Longitude = 0.137163 [WGS84] in seven sites that were representative of a range of typical public dog-walking spaces. These comprised predominantly paved ‘street’ habitats (residential street and seaside promenade); amenity greenspaces (urban parks); and natural greenspaces on the urban-rural interface (Figure 1). Sites were spaced at least 1 km apart to reduce pseudo-replication, based on the mean walking distance the majority of people travel to access amenity and natural greenspace from a local survey [36]. The broad habitats at each site were recorded prior to the formal surveys and grouped into four categories comprising amenity grassland; natural grassland (tall ruderal/rank vegetation, semi-improved grassland, chalk grassland); woodland/scrub and hardstanding (paved or bare ground).

### 2.2. Field Methods

Field surveys were conducted between 1st September and 5th November 2020, to coincide with the highest seasonal risk to dogs from accumulated infection and larval development in gastropods [14,21]. A TinyTag Plus 2 datalogger (NHBS, Ford Road Totnes, UK) was used to collect minimum temperature (°C) and relative humidity (%) at 15-minute intervals at each site. Each logger was deployed at ground level within or on the edge of the sites’ predominant habitat(s) and left in situ for the duration of the study. Surveys of dogs and gastropods were undertaken along a pre-defined 3 km transect route that traversed habitats and on/off footpath locations accessible to dog walkers at each site. The length of transect mirrored the 35-minute period (based on an average walking speed of 5 km/h) reported in the literature as the median length of a daily dog walk in the UK (median 248 min per seven days [37]). The transects were walked by three observers (Baker, R., King, P., Tolhurst, B.) during each of three time periods that were deemed to capture the main activity peaks of both gastropods and dog walkers: morning (06:30–10:30); day (12:00–16:00); and evening (18:30–22:30). During each survey, gastropods were sampled using a 2 m × 2 m portable quadrat which was placed at 20 pre-defined locations situated at approximately 150 m intervals along each transect route. Observers arriving at a quadrat location recorded the time, and for initial surveys the geolocation, predominant habitat category and whether the quadrat was on or off (>2 m from) a footpath. The entire quadrat was then systematically searched, and all gastropods encountered were counted and identified to species where possible, and otherwise to aggregate or genus. Dogs were surveyed using 5-minute point counts that were undertaken at 30-minute intervals during the same sampling periods as the gastropod surveys. At each point count, the location, time, and habitats visible to the observer were logged and the number of dogs per dog walker, habitat category in which each dog was first observed, and whether the dog was on or off (>2 m from) a footpath were recorded. All data were collected using ArcGIS Survey123 (ESRI, Redlands, CA, USA). Surveys were rotated between sample site, time-period, transect and survey week, selected at random.

### 2.3. Data Analysis

#### 2.3.1. Gastropod Generalized Linear Models

All analyses were computed in R v 3.6.1 (The R Foundation for Statistical Computing 2019). Patterns of gastropod presence/absence structured by spatiotemporal and environmental variables were investigated using binary logistic regression with a logit link function, within a generalized linear modelling (GLM) framework. The response variable–gastropod (0,1) was initially regressed against six separate explanatory variables in bivariate analyses: Sample Day; minimum temperature [Mintemp]; minimum relative humidity [Minrh]; Time; Habitat; and location on or off a footpath [Footpath] (Table 1). Correlations between continuous explanatory variables were investigated using Spearman’s Rank correlation as one of these (Sample Day) did not follow a Gaussian distribution. A log-transformed offset of the proportion of each site covered by each habitat type was included to control for variation in available habitat at each site allowing assessment of gastropod habitat use proportional to availability. A final parsimonious model was constructed using forward selection by incrementally adding variables that were significant at the 95% level in bivariate analyses to a multiple regression model based on decreasing deviance values. It was not deemed necessary to adopt additional model selection procedures (e.g., based on AIC) given that the purpose of the GLMs was to screen broad relationships between variables for further refinement in conditional probability trees.

Patterns of gastropod relative abundance structured by spatiotemporal variables were investigated using various zero-inflated models for count data following [38], as the response variable—the number of gastropods in each quadrat, was highly zero-inflated (52% of the response values were zeros). Two candidate distributions were possible fits: zero-inflated Poisson (ZIP) and zero-inflated negative binomial (ZINB). To compare the fit of each of ZIP and ZINB models, each model was generated and a likelihood ratio (LR) test, using the package lmtest, was computed to compare them. This showed that a ZINB was the most appropriate test for our data (Log likelihood ratio, ZINB versus ZIP: −750.38 vs. −1339.33, χ^2^ = 1177.9, *p* « 0.001, “«”means a lot less than). The response was then separately regressed against the same five explanatory variables as for gastropod presence/absence plus a log-transformed habitat offset, and a final parsimonious model constructed using forward model selection. As zero-inflated models do not generate deviance and associated *p* values for each variable overall, LR tests between the variable and the null model in each case were used to produce equivalent values. Chi^2^ and *p* values from LR tests were then used to build the multivariate model by forward selection in place of deviance.

To predict the probability of occurrence of gastropods associated with high and medium levels of *A. vasorum* prevalence, taken to be between 3 and 15%, we created a data subset to include only gastropod taxa for which there was recorded prevalence within those bounds from the literature. This subset contained gastropods from the genera *Tandonia, Derocerus* and *Cornu,* and the *Arion* aggregates *A. hortensis* and *A. ater* [9,13,14,15,21,39]. The presence/absence of these taxa were then regressed against the six explanatory variables as in previous models using identical procedures.

#### 2.3.2. Dog Generalized Linear Models

As for gastropods, patterns of dog occurrence were investigated using binary logistic regression for the analysis with dog presence/absence as the response, and a ZIP model where the response was dog relative abundance (Log likelihood ratio, ZINB versus ZIP = −397.67 vs. 333.91, χ^2^ = 127.51, *p* « 0.001). In each case, we used an identical set of explanatory variables and an identical model selection procedure to the gastropod analyses.

#### 2.3.3. Conditional Probability Trees

We constructed a series of conditional probability (CP) trees in-order to explore the magnitude of difference for the spatiotemporal and environmental patterns detected by the GLM analyses (see Figure 2). For each tree, we considered the significant *n* explanatory variables (e.g., Time, Habitat, etc.) from the GLMs as events E_1_–E_4_. The CPs of observing gastropods or dogs (E_n_) were then calculated by multiplying the observed frequency (probabilities [Pr]) of prior events in the dataset as follows:(1)$$\mathrm{Pr}\left(E_{1}\cap \;E_{2}\;\cap \;E_{3}\ldots \;\cap E_{n}\right)=\mathrm{Pr}\left(E_{1})\mathrm{Pr}(E_{2}|E_{1}\right)\mathrm{Pr}(E_{3}|E_{1}\cap E_{2})\ldots \mathrm{Pr}(E_{n}|E_{1}\cap E_{2}\cap \ldots \;\cap E_{n-1})$$

Conditional probability describes the probability of observing dogs or gastropods given a set of events. Each event has a set of discrete outcomes (such as Habitat classes in our data) and each outcome represents a subset of the data from which the frequencies for the proceeding event outcomes are calculated (e.g., Footpath). We structured our data in order of Time, Minrh, Habitat and Footpath with gastropod or dog presence/absence, or relative abundance, forming the terminal nodes. As Minrh and relative abundance were continuous variables, we first converted these into discrete classes that reflected the distribution of the data. This resulted in three categories for Minrh (low humidity = 40–70%; medium humidity = 71–90%; and high humidity = 91–100%), and four categories for gastropod relative abundance (None = 0 individuals; Low = 1–5 individuals; Medium = 6–20 individuals; High = 21–60 individuals), and dog relative abundance (None = 0 dogs; Low = 0–10 dogs; Medium = 11–18 dogs; High > 18 dogs). A tree that included all four significant explanatory variables in our data would result in the frequencies of observing gastropods or dogs (as presence/absence or across abundance categories) on- or off-footpath for each of the four habitat classes under each of the Minrh categories for each of the Time periods (AM, DAY, PM).

We generated four separate trees, quantifying, as follows: (1) the probability of observing at least one gastropod *given* habitat; (2) the probability of observing at least one dog *given* Footpath, Habitat, Minrh and Time; (3) the probability of observing high, medium and low numbers of gastropods *given* Habitat; (4) the probability of observing at least one gastropod associated with medium or high prevalence of *A. vasorum* infection *given* Habitat, Minrh and Time. We omitted the CP tree for relative abundance of dogs as this was indistinguishable from the dog presence/absence tree due to a low proportion of values in medium and high classes. Hence, only comparisons where dogs were absent (*y* = 0) were frequent enough to yield meaningful results.

We used likelihood ratios to compare the magnitude of the difference between CPs under different conditions in which dogs, gastropods, and gastropods associated with moderate or high prevalence of *A. vasorum* occurred. Any differences over either a magnitude of 2 (i.e., CP twice as high or twice as low) or 10 (10× as high/low), depending on the ratios for each tree, were extracted and expressed as means. We used this to identify the conditions under which gastropods and dogs were most likely to co-occur [P(co)] and for gastropods associated with medium or high prevalence of *A. vasorum*, the conditions under which co-occurrence posed a higher risk of infection [P(inf)]. Lastly, to confirm co-occurrence [P(co)] we performed a correlation analysis using the CPs from the presence/absence trees for both gastropods and dogs given matched conditions. As the CPs did not follow a Gaussian distribution with homogeneity of variance for either dogs or gastropods, the correlation analyses were computed using Spearman’s rank correlation. All trees were computed in Excel and R v 3.6.1. Because multiple quadrat-sized areas were visited by dogs on any given walk, the CPs do not predict actual chance of dog-gastropod encounters, nor infection risk. However, on the basis that these encounters are more likely where both gastropods and dogs are abundant, P(co) is assumed to scale to relative risk of dogs encountering gastropods, and therefore *A. vasorum* exposure, as a function of the variables included in the analysis.

## 3. Results

We recorded a total of 763 domestic dogs and 1672 gastropods (of which 1260 were adults, 331 were juveniles and for the remaining 81 it was not possible to assign age class) over the 2 month period across the seven sites. Overall, the seven commonest gastropod species were: *Cochlicella acuta* (*n* = 548); *Candidula intersecta* (*n* = 211); *C. aspersum* (*n* = 134); *Monacha cantiana* (*n* = 98); *A. hortensis* aggregate (*n* = 98); *Deroceras reticulatum* (*n* = 92); and *Trochulus striolata* (*n* = 90). Overall, Mintemp ranged from 2.75 to 27.27 with a mean of 12.75 [±0.17], and Minrh ranged from 41.54 to 100, with a mean of 90.33 [±0.47]. Sample day was negatively correlated with Mintemp and positively correlated with Minrh (Spearman’s correlation; Mintemp: *rho* = −0.66, *p* « 0.001; Minrh: *rho* = 0.65, *p* « 0.001) indicating that the weather became colder and wetter as the study period progressed.

### 3.1. Gastropod GLMs

Of the initial set of six candidate explanatory variables, only Habitat was significantly correlated with gastropod presence/absence (0,1) at the 95% confidence level in bivariate analyses (F_3,386_ = 129.87, *p* « 0.001). Gastropods were more likely to be present in natural.

Grassland and woodland/scrub relative to amenity grassland and hardstanding habitats (Appendix A). Gastropod relative abundance was highly variable, ranging from 0 to 60 gastropods per quadrat with a mean (±SD) per quadrat of 3.87 ± 8.18. Again, only Habitat was significantly correlated with gastropod abundance at the 95% confidence level in bivariate analyses (GLM, likelihood ratio (bivariate versus null model = 55.65; χ^2^ = 117.3; df = 9, 3; *p* « 0.001)). Gastropods were observed in greater numbers in woodland/scrub versus natural grassland, hardstanding surfaces and amenity grassland (Appendix A). Taxa in the subset of data with medium or high *A. vasorum* prevalence from the literature included slugs of the *A. ater, A. hortensis* aggregates, and of the genera *Derocerus*, *Tandonia*, and the snail *Cornu aspersum*. In bivariate analyses for these taxa, each of Habitat, Sample Day, Time, and Minrh were separately significantly correlated with gastropod presence/absence (0,1) (Habitat: F_3,401_ = 63.23; *p* « 0.001; Sample Day: F_1,403_ = 4.34; *p* < 0.05; Time: F_2,402_ = 12.85; *p* < 0.01; Minrh: F_1,402_ = 6.36; *p* < 0.05). However, Sample Day became non-significant following forwards selection (F_1,396_ = 1.17, *p* > 0.05) leaving three significant predictors in the final model (Table 2). Gastropods of the taxa listed were more likely to be present in the evening than the morning or at midday, and in natural grassland and woodland/scrub relative to amenity grassland and hardstanding habitats (Appendix A). They were also more likely to be encountered in woodland/scrub than natural grassland (Appendix A).

### 3.2. Dog GLMs

Of the 509 dog walker observations from point counts, 42% (214) included more than one dog. Significant explanatory variables predicting presence/absence in bivariate models included Time, Habitat, Minrh and Footpath. All of these were retained in the final model following forward selection, but the effect of Minrh approached significance only and comprised a low relative% deviance (Table 3). The probability of encountering at least one dog (1 versus 0) was higher on a footpath, in hardstanding and amenity grassland versus all other habitats, and in natural grassland versus woodland/scrub (Appendix A). Dog presence was additionally negatively related to Minrh, but this association approached significance only (Appendix A). Dog relative abundance was variable, ranging from 0 to 28 dogs per point count with a mean (±SD) of 1.43 ± 2.68. Significant explanatory variables predicting abundance in bivariate models included Habitat, Minrh (%), Time, and Footpath. The final model retained all four variables, as LR tests showed that the model containing each additional variable was always significantly different to the null model (Table 4). Dog abundance was negatively related to humidity and was greater on a footpath and in amenity grassland and hardstanding habitats versus natural grassland (Appendix A). However, higher numbers of dogs were observed in woodland/scrub relative to natural grassland, following a similar pattern to gastropods (Appendix A).

### 3.3. Gastropod Conditional Probability Trees

The mean likelihood of gastropod presence (1 versus 0) was 4.17× greater in woodland/scrub and natural grassland than on hardstanding surfaces. Further, the mean likelihood of encountering medium or high numbers of gastropods was 5.82× more likely in woodland/scrub or natural grassland than hardstanding and amenity grassland (Figure 3). The mean likelihood of gastropod absence (0 versus 1) was 3.72× greater in amenity grassland and hardstanding surfaces relative to woodland and natural grassland (Figure 3). For the *A. hortensis/A. ater* aggregates, *Derocerus*, *Tandonia* and *C. aspersum* data, unequal comparisons between the different times of day meant that LRs were pooled across the time periods. For this group, the mean likelihood of presence was 59.07× greater in conditions of high humidity, 65.12× greater in woodland and 62.16× greater in amenity grassland than in natural grassland (Figure 4). However, likelihood of presence was also 40.78× more likely on hardstanding surfaces than in natural grassland. Likelihood of absence was greater in amenity grassland versus natural grassland and woodland and at high humidity, but the magnitude was lower (Figure 4).

### 3.4. Dog Conditional Probability Trees

In the morning, dog presence was 14.20× greater on a footpath and 15.57× greater on hardstanding surfaces than all other habitats, and 12.22× greater in natural grassland relative to amenity grassland and woodland/scrub (Figure 5a). However, dog absence was also 16.29× more likely in woodland/scrub than amenity grassland (Figure 5a). At midday, presence was 13.34× more likely on hardstanding surfaces than any other habitat, and absence was 19.61× more likely in woodland/scrub and 14.55× more likely off a footpath (Figure 5). The low occurrence of dog walkers in the evening prevented meaningful LR comparisons for this time of day.

### 3.5. Co-Occurrence of Dogs and Gastropods

A positive correlation was detected in the morning and during the day between the CPs for dogs and gastropods, where these were matched for all other spatiotemporal and environmental conditions, i.e., these were identical in each case (Figure 6).

## 4. Discussion

Our study is the first to simultaneously document patterns of dog walking in public spaces and the occurrence and distribution of common terrestrial gastropod molluscs known to act as intermediate hosts of *A. vasorum*. Our data are instrumental in determining the degree of overlap in situ between final (dog) and intermediate (gastropod) hosts in the *A. vasorum* system according to spatiotemporal and environmental variables. Our findings suggest that the overall presence and relative abundance of gastropods during the study period was uniform between the time periods surveyed and across temperature and relative humidity gradients. Contrastingly, dogs were more likely to be present, and occur at higher relative abundance, in the morning and during the day, at lower humidity levels and on a footpath. Both gastropod and dog occurrence was strongly structured by habitat, yet this diverged to an extent, with gastropods occurring predominantly in natural grassland and woodland/scrub, and dogs occurring predominantly in amenity grassland and on hardstanding surfaces. However, gastropod taxa observed to host high and medium *A. vasorum* prevalence from previous literature (*A. hortensis* and *A. ater* aggregates, *Tandonia*, *Derocerus* and *C. aspersum*) were highly likely to be found in amenity grassland, additional to natural grassland and woodland/scrub. Therefore, we suggest that there is a greater risk to dogs of encountering these taxa than gastropods overall due to a greater degree of habitat use overlap. Nonetheless, there also appeared to be some temporal and humidity-related partitioning, as the higher risk gastropods were more likely to be present in the evening and at high humidity levels, which contrasts with dog activity. When controlling for all other conditions, occurrence of dogs and gastropods were positively correlated in the morning and at midday. Therefore, we detected co-occurrence due to sufficient overlap between habitat, environmental and temporal associations.

Taken together, our findings suggest that habitat is the single most important factor affecting the probability of dogs encountering one or several gastropods, with the likelihood increasing in natural habitats, especially woodland and amenity and natural grassland. Gastropod apparent selection for complex vegetated habitats is likely to stem from a combination of factors including diet, prevention of desiccation and ease of locomotion. Few studies have investigated the impact of substrate on gastropod locomotion. However, one of the most commonly recorded species in this study, *C. aspersum,* has been recorded to move differently on rough and porous surfaces, relative to smooth ones. On rough surfaces (e.g., brick, wood, concrete), this species switches from adhesive crawling to a loping gait and produces slime more intermittently on rough surfaces [40], with potential impacts on energy conservation. Of the species aggregates and genera observed in the literature to occur at high and medium *A. vasorum* prevalence, the majority are either omnivorous or carrion-eaters (e.g., *A. hortensis* and *A. ater* aggregates, *Deroceras*) and include the coprophilic slug *A. hortensis* and coprophilic snail *C. aspersum.*

There are several caveats to the inferences from this study. Firstly, the large variation in gastropod abundance across quadrats was not entirely explained by habitat type, relative humidity, or time of day, which likely relates to intrinsic factors of gastropod populations, such as social aggregation. The strong over-dispersion detected in gastropod abundance is consistent with literature on aggregation in terrestrial gastropod molluscs (e.g., *Derocerus reticulatum,* [41]) which have been shown to follow each other’s slime trails, for example, during homing or reproductive behavior (observed in >30 genera including the coprophilic UK snail *Cepaea nemoralis* [42]). Such non-environmentally forced clustering is likely to explain the non-uniform distribution of gastropods across dog-walking areas, even when habitat types and levels of humidity were suitable for gastropod activity. Secondly, the paucity of large comprehensive datasets, across a broad geographical scale on *A. vasorum* prevalence in different gastropod taxa, means that we can only have limited confidence in the estimation of infection given co-occurrence. Thirdly, we conducted the surveys during only one season (Autumn), which limited the range of temperature and humidity values recorded. Thus, the lack of an effect of relative humidity on gastropod occurrence overall may have resulted from the majority of measurements being within the optimal threshold for which molluscs can maintain homeostasis. However, we specifically conducted the study at the time of year when gastropods were more likely to be active as a direct consequence of suitable prevailing environmental conditions, i.e., high relative humidity but moderate temperature. Further, infection in dogs is highest in Winter/early Spring such that the greatest exposure occurs in late Summer/Autumn [8,43]. This is consistent with accumulated mature *A. vasorum* infections in semelparous slug species [21]) hence, we conducted our study at the riskiest time of year for infection and therefore the most pertinent season for investigating dog and gastropod co-occurrence.

Further study should investigate correlates of dog behavior that affect how likely dogs are to interact with gastropods. These potentially include age and breed. Age may be an important predictor of exploratory behavior as regards investigating gastropods, with infection more likely in puppies and dogs of less than 18 months of age than older dogs [43]. Similarly, some breeds have been associated with a higher risk of infection than others [5] although results are inconsistent between studies. We recommend examination of associations between dog fecal deposition and presence/relative abundance of coprophilic gastropods, and quantification of gastropod aggregation in future studies. Lastly, surveillance of actual *A. vasorum* prevalence in gastropod populations structured by habitat, would allow a deeper level of insight into infection dynamics.

## 5. Conclusions

Our findings indicate that the probability of co-occurrence *P(co)* between domestic dogs and gastropod molluscs overall in public dog walking spaces strongly increases in complex vegetated (natural) environments, and strongly decreases on artificial, hardstanding surfaces. These associations persist for gastropod taxa at high and medium *A. vasorum* infection risk, but for these taxa, the likelihood of presence additionally increases in the evening, at high humidity and in managed (amenity) environments. We recommend future work investigating the dynamics of coprophilic gastropod behavior, dog behavior, and dog fecal deposition in high-risk habitats, alongside generation of simultaneous data on prevalence in populations of both host types. This will inform further predictive models of the probability of infection [*P(inf*)].

## Figures and Tables

**Figure 1 animals-11-02577-f001:**
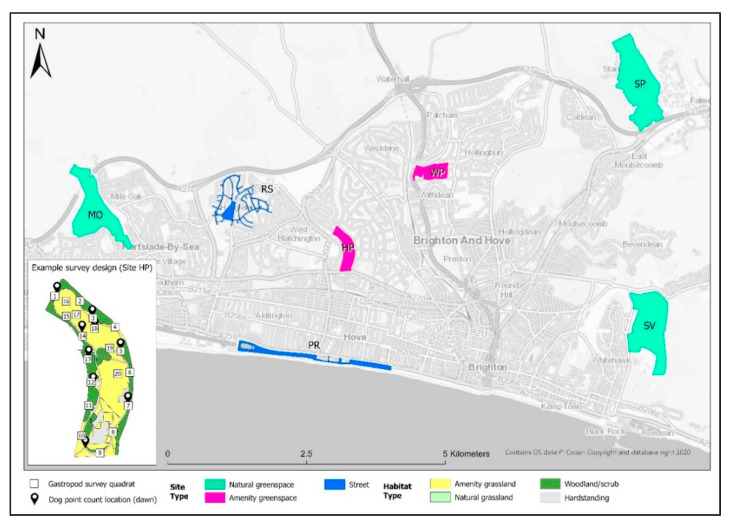
Field sites across the city of Brighton and Hove, England, UK and surrounding areas where quadrat and point count transect surveys were conducted for gastropod (mollusca) and dog (*Canis lupus familiaris*) occurrence under different spatiotemporal and environmental conditions.

**Figure 2 animals-11-02577-f002:**
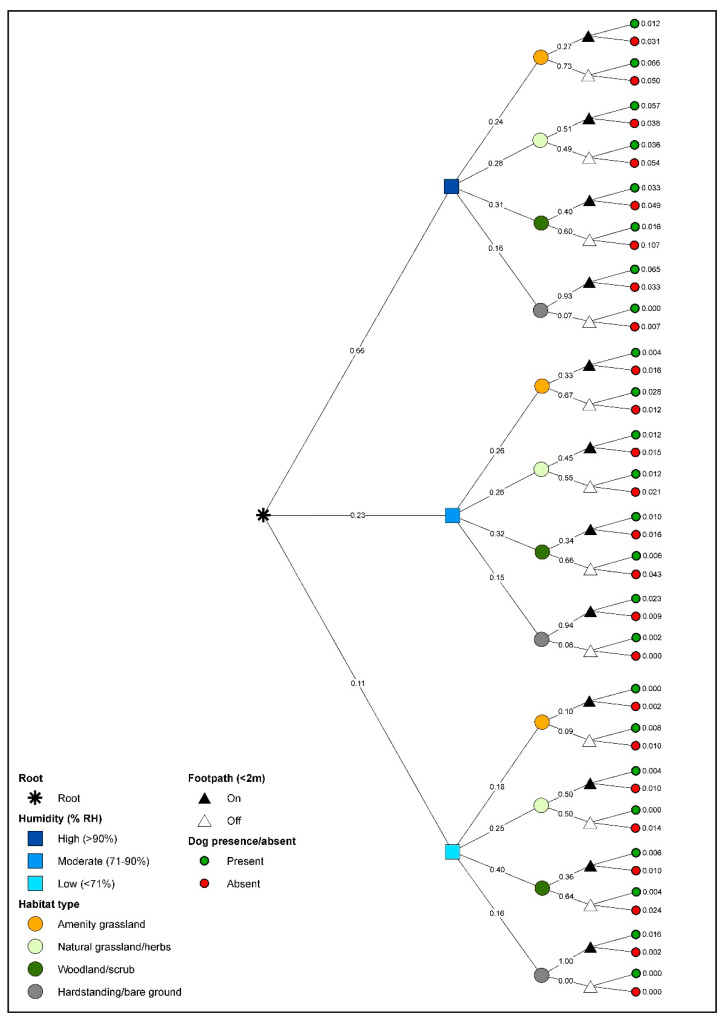
Example of conditional probability tree for determining differences between likelihood of encounter, in this case for the presence/absence of dogs (*Canis lupus familiaris*) across 7 sites in Brighton and Hove and surrounding areas, England, UK.

**Figure 3 animals-11-02577-f003:**
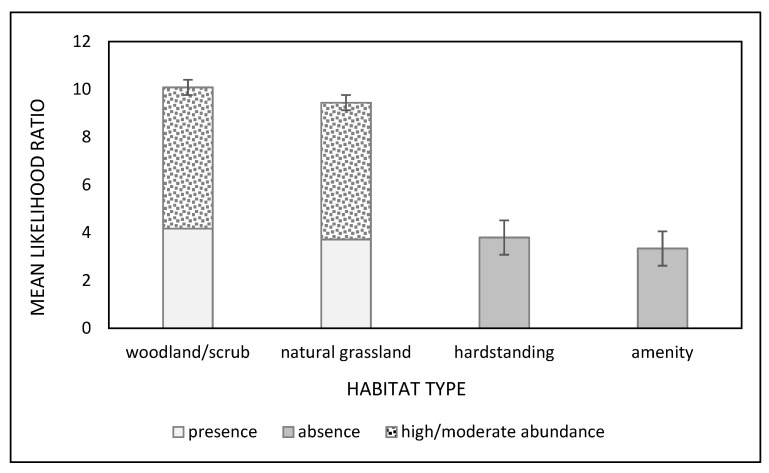
Mean likelihood ratios ± standard error, for presence and absence, and moderate and high relative abundance of all gastropods. Data were derived from quadrat data across 7 study sites in Brighton and Hove and surrounding areas, England, UK.

**Figure 4 animals-11-02577-f004:**
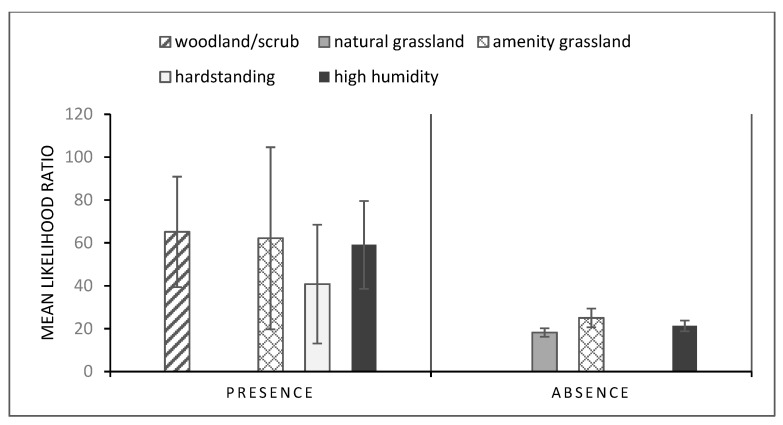
Mean likelihood ratios ± standard error, where conditional probabilities were >10× magnitude for presence and absence across the three time periods of gastropods from the *Arion ater* and *Arion hortensis* aggregates, *Tandonia*, *Derocerus* and *Cornu aspersum* (high and medium prevalence taxa for *Angiostrongylus vasorum* from the literature). Data were derived from quadrat data across 7 study sites in Brighton and Hove and surrounding areas, England, UK.

**Figure 5 animals-11-02577-f005:**
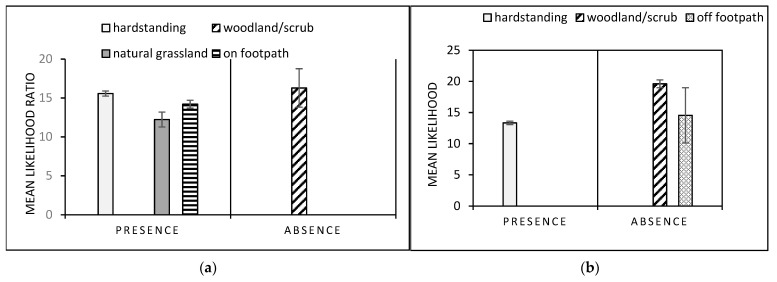
Mean likelihood ratios, ±standard error, from conditional probability trees where conditional probabilities were >10 times magnitude for dog (*Canis lupus familiaris*) presence and absence during the: (**a**) morning (06.30–10.30) and (**b**) day (12.30–16.00). Data were derived from point count transect data across 7 study sites in Brighton and Hove and surrounding areas, England, UK.

**Figure 6 animals-11-02577-f006:**
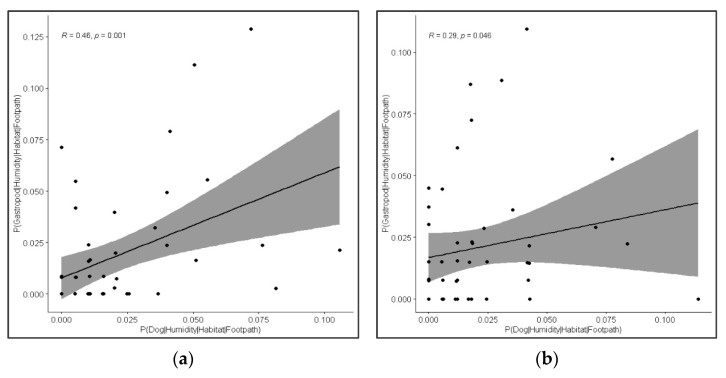
Correlation between conditional probabilities for dogs (*Canis lupus familiaris*) and for gastropods (mollusca) during the; (**a**) morning, and (**b**) day, where all other conditions were matched. Data were derived from quadrat and point count transect data across 7 study sites in Brighton and Hove and surrounding areas, England, UK.

**Table 1 animals-11-02577-t001:** Explanatory variables predicting each of gastropod presence, gastropod relative abundance, dog presence and dog relative abundance in Generalized Linear Models (GLMs) at seven sites in Brighton and Hove and surrounding areas in England, UK.

Variable Name	Variable Description	Variable Type	Levels
Sample Day	1–21 progressive sampling days	Continuous	n/a
Mintemp	minimum temperature (°C) measured in a central location at the site, recorded at 15-minute intervals	Continuous	n/a
Minrh	Minimum relative humidity (%) in central location recorded at 15-minute intervals	Continuous	n/a
Time	Time of day between 06.00 and 23.00	Nominal	AM (06.30–10.30) DAY (12.30–16.30) PM (18.30–23.30)
Habitat	Classification of quadrat habitat type	Nominal	hardstanding (bare ground and artificial substrates, e.g., paved) amenity grassland natural grassland woodland/scrub
Footpath	Location of quadrat on or off (<2 m from) a footpath	Nominal	off footpath on footpath

**Table 2 animals-11-02577-t002:** Model effects for explanatory variables predicting the presence/absence of *Arion hortensis* and *Arion ater* aggregates, and *Cornu aspersum* (taxa harboring medium and high prevalence of *Angiostrongylus vasorum* from the literature) from quadrats along line transects across 7 study sites in Brighton and Hove and surrounding areas, England, UK. Habitat = habitat type (hardstanding, amenity grassland, natural grassland, woodland/scrub); Minrh = minimum relative humidity (%); Time = time of day (morning, midday, evening). Total deviance = the difference between null (457.13 on 404 *df*) and residual (365.54 on 397 *df*) deviance for model overall = 92. * < 0.05; ** < 0.01; *** < 0.001.

Model Term	Deviance	*df*	*%* Deviance ([Term Deviance/Total Deviance] × 100)	*p*
Habitat	73.35	396	79	<0.001 ***
Time	12.14	396	13	<0.01 **
Minrh	7.76	397	8	<0.05 *

**Table 3 animals-11-02577-t003:** Model effects for explanatory variables predicting dog (*Canis lupus familiaris*) presence/absence from point count transect data across 7 study sites in Brighton and Hove and surrounding areas, England, UK. Time = time of day (morning, day, evening); Habitat = habitat type (hardstanding, amenity grassland, natural grassland, woodland/scrub); Minrh = minimum relative humidity (%); Footpath = location on/off a footpath. Total deviance = the difference between null (660.02 on 507 df) and residual (536.46 on 500 df) deviance for model overall = 123.74. *** < 0.001.

Model Term	Deviance	*df*	*%* Deviance ([Term Deviance/Total Deviance] × 100)	*p*
Time	67.279	501	54	<0.001 ***
Habitat	38.563	503	31	<0.001 ***
Footpath	14.153	501	11	<0.001 ***
Minrh	3.569	500	3	0.059

**Table 4 animals-11-02577-t004:** Likelihood Ratio (LR) tests predicting dog (*Canis lupus familiaris*) relative abundance across 7 study sites in Brighton and Hove and surrounding areas, England, UK. Each LR comparison and ꭕ^2^ value is for the model listed in each case against the previous model in the row above except for Time only, which is tested against the null model (dog~1). Time = time of day (morning, day, evening); Habitat = habitat type (hardstanding, amenity grassland, natural grassland, woodland/scrub); Footpath = location on/off a footpath; Minrh = minimum relative humidity (%). * < 0.05; ** < 0.01; *** < 0.001.

Model	LR Difference	*df*	*ꭕ^2^*	*p*
Time	−756.15 vs. 786.01	4	59.717	<0.001 ***
Time + Habitat	−739.25 vs. −756.15	6	33.791	<0.001 ***
Time + Habitat + Footpath	−735.55 vs. −739.25	2	7.402	<0.05 *
Time + Habitat + Footpath + Minrh	−730.68 vs. −739.25	2	9.750	<0.01 **

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
