# Peer review of "Co-Occurrence of Domestic Dogs and Gastropod Molluscs in Public Dog-Walking Spaces and Implications for Infection with Angiostrongylus vasorum: A Preliminary Study"

_animals, 2021, doi:10.3390/ani11092577_

Round 1

Reviewer 1 Report

The present study provides data about co-occurrence of domestic dogs and gastropod molluscs in public dog-walking spaces and implications for infection with Angiostrongylus vasorum. Authors showed that the co-occurrence (Pco) between domestic dogs and gastropod molluscs overall in public dog walking spaces strongly increases in complex vegetated (natural) environments, and strongly decreases on artificial, hard standing surfaces. Despite the interest of the topic, my major concern and the major follow in the present work is about the design of the study since there are several caveats (mentioned by the authors themselves).  Firstly and most importantly, the study was performed during the period from September to November which is only one season (two months). I understand that this duration could be the most appropriate season for the research question. However, this period is not sufficient to have a general idea about the studied topic. Furthermore, the paucity of large comprehensive datasets across a broad geographical scale and gastropod abundance across quadrats were not entirely explained by habitat, humidity or time of day. All these caveats mentioned render the manuscript to be accepted as they represent a major flows and I believe additional experiments are needed during longer period to involve more seasons as the current study can't give us complete idea about the topic.

Other comments:

English language style requires moderate editing as there is several typos errors. Revising the simple summary, the authors repeated ‘’ we‘’ several times, I guess it is better to be formal and in science, ‘’we‘’ should be avoided as much as possible.

Introduction contains too much information, some of which are repetitive or less important, making distraction of the reads

Journal style and author guidelines should be followed and ethical approval code should be provided.

My suggestion is to reject the manuscript due to these serious flaws since additional experiments are needed to cover all the seasons. Once these major flaws are covered, the manuscript would deserve publishing in good journal like animals.

Reviewer 2 Report

This is an interesting ms investigating the spacio-temporal pattern of occurrence and co-occurrence of slugs & snails and dogs & dog walkers in urban and suburban areas where the risk of lungworm infection in dogs is of significance.

Main comment

Not clear description of the results. Although the authors put a lot of effort in analysis and presentation of the results, I found few parts of result section hard to follow and even confusing. This concerns mainly the results described by the authors themselves as ‘imprecise’ or of ‘low confidence’. I would suggest to either delete this part (if its significance is doubtful) or eventually mentioning about it in discussion. Data presented on Figure 4, including columns for ‘absence’, are not clear. The authors again describe this fig as ‘imprecise or with ‘large errors’ (?) (lines 395-400).

I would generally suggest to simplify and clarify the presentation of the results (especially lines 395-400), avoiding presentation of inconclusive/imprecise or erroneous data.

Figures 3, 4 and 5 are hard to understand (columns for absence- also their description is hard to follow)

Minor comments

Authors introduced : zero-inflated Poisson (ZIP) and zero-inflated negative binomial (ZINB); correct lower letters in lines 218-219.

The use of ‘av’ and full Latin names: the authors should follow general rules for presentation of Latin names of organisms: full Latin name shall be presented at the first use; then the first part shall be abbreviated: full names of snails/slugs are used through the text and this shall be re-formatted. In contrast, the authors use ‘av’ instead of A. vasorum; I would suggest the use of this form instead of ‘av’ through the text.

Beside, ‘lungworm’ is an accepted term (French worm is colloquial), so it can be used.

Reviewer 3 Report

I found this manuscript as an interesting one and actually I never read such articles before on predicting the prevalence of a dog parasite by analyzing the intermediate host availability and different habitat types.

In methods, dogs and gastropods were observed and analyzed the co-occurrence, but it would me more nice if the surveyed dogs were also tested for Angiostrongylus vasorum infection. Then it would be possible to assume the real prevalence of this lungworm infection and also to analyze the co-occurrence of dogs and  intermediate hosts in study areas.

The authors is suggested to correct the points raised in the attached reviewed file.

Author Response

Response to Reviewer 3

Many thanks for all the suggested changes on the pdf, which were very helpful and improved the MS. All changes have been made and are shown highlighted yellow in the MS text. Specific changes of note are:

Line 54 ‘patchily’ changed to ‘unevenly’ and UK changed to United Kingdom on first mention and UK thereafter

Av changed to lungworm throughout

Line 309 ‘unknown’ gastropods is now qualified as (not possible to assign age class)

Round 2

Reviewer 1 Report

The authors thank the provided a revised version of the manuscript after addressing some comments and including some changes. However, the major flaws still exist and my major concern is that the research was no conducted correctly (the study duration in particular). Authors mentioned that the study period (two months) might be the most appropriate season but this period is not sufficient to have a general idea about the studied topic and for addressing the research. Authors mentioned in their reply that '' It is our sincere hope that our results will inspire others to conduct additional studies ''. However, I am not convinced by their response since the study design is very critical and research should be based on clear articulated design and realistic deliverables rather than hopes and inspirations. Furthermore, gastropod abundance across quadrats were not entirely explained by habitat, humidity or time of day and the paucity of large comprehensive data sets across a broad geographical scale. All these caveats represent serious flaws and I believe additional experiments are needed in order to cover more seasons and to give us complete idea about the topic. Therefore, these serious flaws render the manuscript to be accepted and my suggestion is to reject the manuscript and encourage its re-submission after doing extra work.

Author Response

We are not aware of any additional comments from Reviewer 1. We attach the revised MS according to minor edits requested by the academic editor, below
